# Beyond Speedup - Utilizing KV Cache for Sampling and Reasoning

**Zeyu Xing**[1], **Xing Li**[2], **Hui-Ling Zhen**[2], **Mingxuan Yuan**[2], **Sinno Jialin Pan**[1]

[1]Department of Computer Science and Engineering, The Chinese University of Hong Kong
[2]Huawei Technologies Co., Ltd.
`zeyuxing@link.cuhk.edu.hk`
`{li.xing2, zhenhuiling2, yuan.mingxuan}@huawei.com`
`sinnopan@cuhk.edu.hk`

## Abstract

KV caches, typically used only to speed up autoregressive decoding, encode contextual information that can be reused for downstream tasks at no extra cost. We propose treating the KV cache as a lightweight representation, eliminating the need to recompute or store full hidden states. Despite being weaker than dedicated embeddings, **KV-Embeddings** are shown to be sufficient for two key applications: **(i) Chain-of-Embedding**, where they achieve competitive or superior performance on Llama-3.1-8B-Instruct and Qwen2-7B-Instruct; and **(ii) Fast/Slow Thinking Switching**, where they enable adaptive reasoning on Qwen3-8B and DeepSeek-R1-Distil-Qwen-14B, reducing token generation by up to $5.7\times$ with minimal accuracy loss. Our findings establish KV caches as a free, effective substrate for sampling and reasoning, opening new directions for representation reuse in LLM inference. **Our code is available at** `https://github.com/cmd2001/ICLR2026_KV-Embedding`.

## 1 Introduction

Large language models (LLMs) rely on key-value (KV) cache to accelerate autoregressive decoding by reusing past attention states, avoiding costly recomputation. This makes the KV cache indispensable for low-latency inference in production systems like vLLM (Kwon et al., 2023). However, its role is typically confined to this speedup. Beyond acceleration, the KV cache is seldom viewed as a reusable representation—with the notable exception of cache steering, a technique that modifies the cache's initial state to guide generation (Belitsky et al., 2025).

While the KV cache has been mostly confined to acceleration (Li et al., 2025; Yang et al., 2025), hidden states have been widely exploited for *self-evaluation* (Wang et al., 2025b; Chen et al., 2024; Beigi et al., 2024; Zhang et al., 2025a) and for *adaptive reasoning and control* (Zhang et al., 2025b; Wang et al., 2025a; Yue et al., 2025). These methods, however, rely on storing full hidden states, which is costly in both memory and computation.

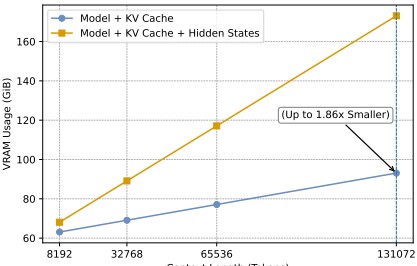

Figure 1: VRAM usage vs. context length for Qwen3-32B (QwenTeam, 2025), comparing Model+KV Cache vs. Model+KV Cache+Hidden States.

In this work, we investigate a simple but powerful question: **Can the KV cache do more than just accelerate decoding?** Since the KV cache is already computed and stored during inference, using it for downstream tasks incurs *no additional cost*. This is a major advantage over storing full hidden states, which is prohibitively expensive in terms of memory. As shown in Figure 1, the KV cache offers a significantly more compact and practical alternative for typical decoder-only models.

Though the KV cache is not explicitly trained as a general-purpose embedding—its sole objective is to support next-token prediction—we find it nonetheless encodes rich contextual information

suitable for various downstream tasks. We explore the potential of **KV-Embeddings** through two applications:

- **Chain-of-Embedding:** We repurpose the KV cache as a lightweight and readily available embedding. In experiments on Chain of Embedding (CoE) (Wang et al., 2025b)—a method for selecting optimal reasoning paths without external information—we show that KV caches achieve classification performance comparable to or even surpassing that of using the model's hidden states.
- **Fast/Slow Thinking Switch:** We leverage the KV cache to implement an adaptive switching mechanism between fast, low-compute reasoning and slower, deliberate reasoning. By reusing KV cache, this approach achieves substantial efficiency gains with minimal performance loss.

Our contributions are fourfold:

1. We present the first systematic study of KV caches as reusable task representations, showing they can be repurposed at near-zero computational cost. In particular, we propose simple but effective aggregation techniques that make KV caches directly usable as embeddings.
2. Despite not being designed as general-purpose embeddings, we find that KV cache representations when processed with the proposed aggregation strategies, are competitively effective on certain classification tasks.
3. We propose **KV-CoE**, a variant of Chain-of-Embedding that reuses the KV cache already stored during decoding. KV-CoE achieves self-evaluation without extra activation storage, offering nearly zero memory overhead and seamless integration into existing inference frameworks.
4. We introduce **KVClassifier**, a fast/slow auto-thinking framework that reuses KV caches for adaptive reasoning with minimal overhead.

Our results suggest that KV caches are a versatile and low-cost foundation for sampling and reasoning, moving beyond their traditional role as a mere acceleration component to become a core resource for effective and efficient LLM-based inference.

## 2 RELATED WORK

**Hidden–state self-evaluation.** A growing line of work shows that internal activations encode reliable signals about answer correctness and hallucination risk. Wang et al. (2025b) propose *Chain-of-Embedding* (CoE), which models the trajectory of layerwise hidden states during inference and derives output-free correctness scores from the geometry of this path. Chen et al. (2024) (INSIDE) introduce *EigenScore*, computed from the eigenspectrum of hidden-state covariance, to assess semantic (in)consistency and detect hallucinations. Beigi et al. (2024) train a contrastive probe on *internal states* (attention, MLP activations) to produce well-calibrated confidence estimates across NLU/NLG tasks. Zhang et al. (2025a) further probes hidden states of reasoning models to predict whether a generated answer will be correct. All of these methods operate directly on hidden states or logits. **Our study**, by contrast, investigates whether *the KV cache alone*—which is already present at inference—suffices to support the same families of subtasks.

**Adaptive fast/slow reasoning and dynamic control.** To mitigate overthinking on easy inputs and underthinking on hard ones, recent work explores *adaptive* reasoning depth (Xing et al., 2025). Zhang et al. (2025b) quantify upper bounds of long- vs. no-thinking modes and propose *Adaptive Self-Recovery Reasoning* (ASRR), adding accuracy-aware length rewards to reduce unnecessary reasoning while allowing implicit recovery. PATS (Wang et al., 2025a) performs *process-level* switching via process reward models with beam search, enabling step-wise fast/slow adaptation with bad-step penalties. DOTS (Yue et al., 2025) views reasoning as a search over atomic actions and learn to select dynamic trajectories. These approaches typically require explicit chain-of-thought generation, external reward models, or re-decoding. **Our contribution** is orthogonal: we show that pooled *KV-cache* features can drive both one-shot (classification-style) and in-generation (generative-style) switching via simple control tokens, without storing hidden states or altering model architecture.

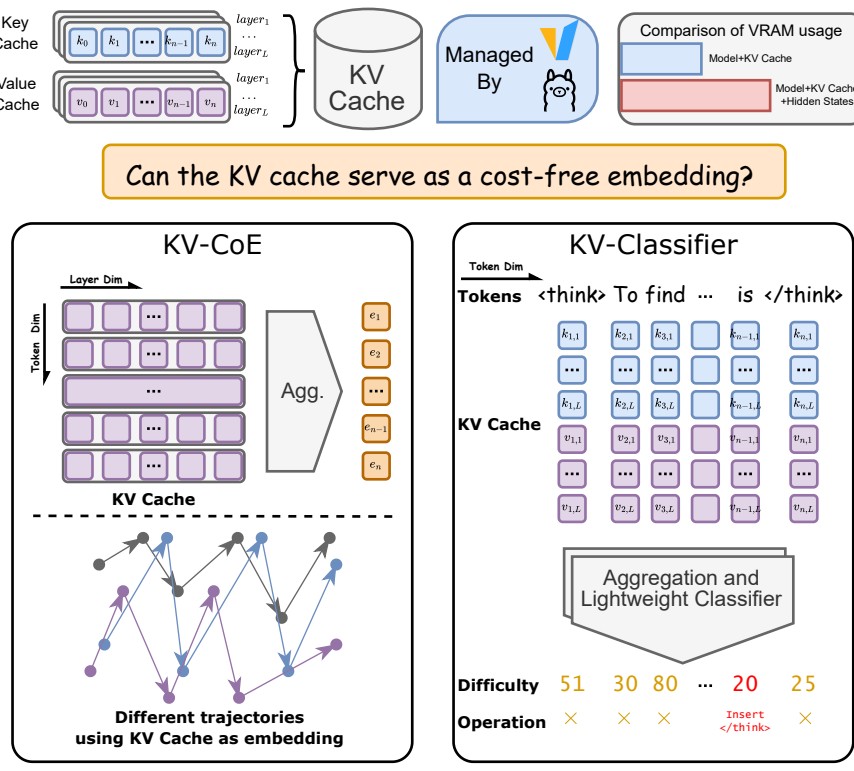

Figure 2: The proposed **KV-Embedding** framework reuses KV cache as a lightweight representation for two applications: (a) **KV-CoE** (Chain-of-Embedding) for self-evaluation of reasoning paths, and (b) **KVClassifier** for fast/slow thinking switch. KV cache, computed during normal inference, is aggregated into embeddings that drive path selection or difficulty-aware switching without storing full hidden states, achieving near-zero overhead.

**KV-cache interventions.** While our work treats the KV cache as a read-only representation for evaluation and control, a concurrent line of research shows that it can also serve as a *control interface*. Belitsky et al. (2025) introduce *KV Cache Steering*, a one-shot intervention that adds layer-wise steering vectors—derived from contrastive CoT vs. non-CoT prompts—to the key and value tensors after prefill, reliably inducing longer and more structured reasoning in small Language Models. Compared with activation steering, this method offers improved stability and negligible runtime overhead. **Our approach** is complementary: instead of modifying the cache, we *pool* it to derive difficulty-aware signals that gate slow reasoning.

## 3 BACKGROUND

### 3.1 TRANSFORMER, HIDDEN STATES, AND KV CACHE

Decoder-only transformers are the architectural foundation of modern large language models (LLMs) (Vaswani et al., 2017; Brown et al., 2020). During autoregressive generation, each transformer layer processes a new token to produce a contextual *hidden state*. A computational bottleneck arises because standard attention requires recomputing over all previous tokens at each step, resulting in $\mathcal{O}(T^2)$ complexity per step, where $T$ is the sequence length. To mitigate this, the key–value (KV) cache (Dao et al., 2022) stores the attention keys and values for all past tokens at every layer. This allows the model to compute keys and values only for the new token and attend to the cached history, reducing the complexity to $\mathcal{O}(T)$ per step and enabling efficient long-sequence generation.

Formally, for layer $l$, we store

$$\text{KVCache}^{(l)} = \{K_{1:T}^{(l)}, V_{1:T}^{(l)}\},$$

where $K_{1:T}^{(l)}, V_{1:T}^{(l)} \in \mathbb{R}^{T \times H \times d_{\text{head}}}$ are the stacked key and value tensors across all attention heads $H$. The hidden state at step $t$ is computed via

$$h_t^{(l)} = \text{Attention}\left(Q_t^{(l)}, K_{1:t}^{(l)}, V_{1:t}^{(l)}\right).$$

Since $K_{1:t-1}^{(l)}$ and $V_{1:t-1}^{(l)}$ are already cached, only $K_t^{(l)}$ and $V_t^{(l)}$ need to be computed online.

## 3.2 MODERN LLM FRAMEWORKS AND KV CACHE MANAGEMENT

State-of-the-art LLM serving frameworks carefully manage KV caches to achieve high throughput, low latency, and efficient GPU memory utilization.

Modern LLM frameworks treat the KV cache as a first-class resource. vLLM (Kwon et al., 2023) introduces *PagedAttention*, virtualizing the KV cache with a paging mechanism akin to CPU virtual memory to enable dynamic allocation, low fragmentation, and high-throughput serving under heavy concurrency. In contrast, Ollama (Ollama Team, 2024) focuses on lightweight, developer-friendly deployment, managing KV caches at the session level to efficiently reuse context across multi-turn interactions. Together, these systems illustrate how both production-scale and interactive inference workloads rely on persistent, reusable KV caches as a core abstraction.

Across these frameworks, the KV cache is managed as a first-class resource with explicit strategies for allocation, eviction, and reuse. This observation motivates our central claim:

"*Since the KV cache is an unavoidable byproduct of efficient inference, repurposing it for downstream tasks adds virtually no overhead.*"

## 4 OBSERVATION

### 4.1 CAN KV CACHES SERVE AS AN EMBEDDING SOURCE

The hidden states and attention projections stored in KV caches encode contextualized token representations, making them natural candidates for use as embeddings. While recent work has explored leveraging intermediate representations from LLMs as task-specific embeddings (Liu et al., 2024), we specifically investigate aggregating KV cache vectors into sentence-level representations.

To evaluate their quality as an embedding source, we construct embeddings by concatenating keys and values at every layer, then averaging across token positions, attention heads, and layers before applying $\ell_2$ normalization. We benchmark these KV-derived embeddings on the Massive Text Embedding Benchmark (MTEB) (Muennighoff et al., 2023) against a strong, dedicated embedding model (gemini-embedding-001).

| Dataset | Llama-3.1-8B-Instruct KV Cache | Gemini-Embedding-001 |
|---|---|---|
| AmazonCounterfactualClassification | 0.3530 | 0.8820 |
| DBpediaClassification | 0.5937 | 0.9476 |
| FinancialPhrasebankClassification | 0.6254 | 0.8864 |
| TweetTopicSingleClassification | 0.3714 | 0.7111 |

Table 1: Performance of KV cache-based embeddings vs. a dedicated embedding model on selected MTEB classification tasks. Despite being significantly weaker than trained embeddings, KV-derived embeddings still capture meaningful semantics. Appendix C further reports hidden-state and chance baselines under the same evaluation pipeline.

As shown in Table 1, KV-derived embeddings significantly underperform their dedicated counterpart across all datasets, confirming they are *not perfect general-purpose embeddings*. This gap stems from three factors: (i) KV representations are optimized for causal language modeling, not contrastive learning, leading to poor isotropy; (ii) they are inherently token- and position-specific, requiring heuristic pooling for sentence-level use; and (iii) their projection into a lower-dimensional head space ($d_{\text{head}} \ll d_{\text{model}}$) reduces their discriminative power.

Despite these limitations, the results show that KV caches encode substantial semantic information—enough to be competitive on certain classification tasks. This finding motivates our exploration of reusing the KV cache for *Chain-of-Embedding* and *Fast/Slow Thinking Switch*, where global embedding quality is less critical than local, relative separability between candidates.

## 4.2 WHY KV CACHES ARE SUFFICIENT FOR CHAIN-OF-EMBEDDING AND FAST/SLOW THINKING SWITCH?

KV caches are poor *general-purpose* embeddings: they are trained for next-token prediction, are position/context dependent, and the space is often anisotropic. Still, they are **sufficient** for our two uses—*Chain-of-Embedding* (CoE) and *Fast/Slow Thinking Switch*—because both rely on *local, task-conditioned* comparisons rather than globally calibrated semantics.

**Local (restricted-set) adequacy.** General embedding learning targets global separation across a broad label/instance space. Here we only need correct *relative ordering* within a small candidate set $\mathcal{C}$ (e.g., a small label space or a few candidate continuations). Concretely, for a decision rule $f$ and score gap

$$\gamma(x) = f_{y_i}(x) - f_{y_j}(x),$$

we only require $\min_{y \in \mathcal{C}} \gamma(y) > 0$, not a globally well-structured embedding space. This is why KV embeddings can work well on *certain classification-like* cases with limited labels/candidates, despite weak MTEB-style semantic similarity.

**CoE is not a classification.** CoE extracts a *path embedding* along the reasoning trajectory and estimates the chance of correction from local geometry (e.g., angle/step length). The requirement is therefore weaker than global separability: KV embeddings only need to preserve these *local trajectory cues* consistently over short ranges.

**Task conditioning + efficiency.** The pooled KV embedding $e = \text{Pool}(K, V)$ is conditioned on input and instruction, $e = g(x, \iota)$, which already biases the representation toward the current task. Finally, reusing KV caches is essentially free compared to storing hidden states ($C_{\text{hidden}} \gg C_{\text{KV}} \approx 0$), so in memory/latency-sensitive regimes the utility favors KV reuse even if accuracy is slightly lower.

**Scope and limitations.** We do *not* claim KV caches yield universally strong embeddings. They are unsuitable when a task needs globally comparable semantics across diverse queries (e.g., broad retrieval/similarity). Our claim is limited to (i) restricted candidate sets and (ii) CoE-style local trajectory geometry.

## 5 CHAIN OF EMBEDDING WITH KV CACHE

LLMs exhibit emergent reasoning capabilities, though their internal decision processes remain largely opaque. To address this, Wang et al. (2025b) introduce *Chain-of-Embedding* (CoE), a method that probes the model's latent space by tracking the evolution of sentence-level representations across layers. Formally, for an LLM $\mathcal{M}$ with $L$ layers, let $h_l^{(t)}$ denote the hidden representation of token $t$ at layer $l$. The sentence-level representation at layer $l$ is obtained by averaging over the sequence length $T$:

$$s_l = \frac{1}{T} \sum_{t=1}^{T} h_l^{(t)}, \quad l = 0, 1, \ldots, L. \tag{1}$$

The CoE trajectory is then defined as the sequence of these layer-wise representations:

$$\text{CoE} = \{s_0, s_1, \ldots, s_L\}. \tag{2}$$

CoE characterizes this trajectory by measuring both magnitude and directional changes of embeddings between consecutive layers:

$$\Delta r_l = \|s_{l+1} - s_l\|_2, \text{ and } \Delta \theta_l = \arccos\left(\frac{s_{l+1} \cdot s_l}{\|s_{l+1}\| \|s_l\|}\right). \tag{3}$$

These features are aggregated into self-evaluation scores. For instance, the real-space combination (CoE-R) is

$$\text{CoE-R} = \frac{1}{L-1} \sum_{l=0}^{L-1} \left( \alpha \Delta r_l + \beta \Delta \theta_l \right), \tag{4}$$

where $\alpha, \beta$ are weighting coefficients. A more robust complex-space variant (CoE-C) treats each $(\Delta r_l, \Delta \theta_l)$ pair as a complex number $z_l = \Delta r_l + i \Delta \theta_l$ and computes the magnitude of their average:

$$\text{CoE-C} = \left| \frac{1}{L-1} \sum_{l=0}^{L-1} z_l \right|. \tag{5}$$

CoE has demonstrated strong discriminative power in distinguishing correct from incorrect model generations, achieving state-of-the-art performance on self-evaluation benchmarks.

## 5.1 METHODOLOGY

Our key innovation is to adapt the CoE framework to use the KV cache, eliminating its primary computational overhead. While vanilla CoE constructs trajectories from hidden states $h_l^{(t)}$, requiring expensive activation storage or re-computation, we instead leverage the key-value tensors $K^{(l,t)}, V^{(l,t)}$ that are already maintained by autoregressive decoders. This modification preserves the CoE analytical framework while rendering it virtually cost-free.

**Embedding Construction.** For each token $t$ and layer $l$, we start with the cached key-value tensors $K^{(l,t)}, V^{(l,t)} \in \mathbb{R}^{H \times d}$. We flatten the head and key/value dimensions and average across layers to produce a compact per-token embedding:

$$e_t = \frac{1}{L} \sum_{l=1}^{L} \text{flatten}\big( K^{(l,t)}, V^{(l,t)} \big) \in \mathbb{R}^{H \cdot d}. \tag{6}$$

The resulting token-wise trajectory is defined as:

$$\text{KV-CoE} = \{e_1, e_2, \ldots, e_T\}, \tag{7}$$

which directly parallels the structure of vanilla CoE but operates along the token dimension.

**Trajectory Characterization.** We characterize this trajectory using the established CoE metrics, simply substituting the token index $t$ for the layer index $l$:

$$\Delta r_t = \|e_{t+1} - e_t\|_2, \text{ and } \Delta \theta_t = \arccos\left( \frac{e_{t+1} \cdot e_t}{\|e_{t+1}\|_2 \|e_t\|_2} \right), \tag{8}$$

$$\text{KV-CoE-R} = \frac{1}{T-1} \sum_{t=1}^{T-1} \left( \alpha \Delta r_t + \beta \Delta \theta_t \right), \text{ and } \text{KV-CoE-C} = \left| \frac{1}{T-1} \sum_{t=1}^{T-1} (\Delta r_t + i \Delta \theta_t) \right|. \tag{9}$$

These formulations maintain the analytical rigor of CoE-R and CoE-C with minimal conceptual alteration.

**Contributions and Advantages.** As illustrated in Figure 3, which compares vanilla CoE and our **KV-CoE**, our method offers two main advantages:

1. **No extra activation cost.** Since the KV cache is already computed and stored during standard autoregressive decoding, reusing it for trajectory analysis incurs virtually no additional activation cost. The required reductions are computationally negligible compared to a full forward pass, resulting in $\Delta M \approx 0$ extra memory and minimal FLOPs.

2. **Deployment-friendly.** The approach works directly with standard inference stacks (e.g., `past_key_values` in Transformers or vLLM). It requires no architectural changes, re-forwarding, or activation hooks, making it immediately deployable in production LLM serving systems.

## 5.2 EXPERIMENTAL RESULTS

| Model | Method | MATH | | TheoremQA | |
|-------|--------|------|------|-----------|------|
| | | AUROC ↑ | FPR95 ↓ | AUROC ↑ | FPR95 ↓ |
| Llama-3.1-8B-Instruct | MaxProb | 59.16 | 87.96 | 45.41 | 98.60 |
| | PPL | 60.82 | 86.42 | 46.45 | 97.82 |
| | Entropy | 62.74 | 84.14 | 47.37 | 97.82 |
| | CoE-R†(Llama3-8B) | 72.54 | 75.61 | 63.12 | 89.83 |
| | CoE-C†(Llama3-8B) | 73.08 | 79.60 | 55.85 | 90.14 |
| | **KV-CoE-R (ours)** | **64.36** | **63.82** | 74.74 | 62.93 |
| | KV-CoE-C (ours) | 64.13 | 67.42 | **74.93** | **62.46** |
| Qwen2-7B-Instruct | MaxProb | 12.40 | 99.34 | 4.92 | 99.87 |
| | PPL | 12.43 | 99.50 | 5.11 | 100.00 |
| | Entropy | 16.19 | 99.42 | 5.28 | 99.87 |
| | CoE-R | 75.75 | 65.95 | 66.68 | 85.84 |
| | CoE-C | 76.68 | 64.48 | 62.70 | 87.42 |
| | KV-CoE-R (ours) | 76.92 | 49.83 | **88.87** | **54.30** |
| | **KV-CoE-C (ours)** | **84.12** | **44.82** | 83.27 | 58.35 |

Table 2: Self-evaluation results on reasoning tasks. KV-CoE consistently improves AUROC and reduces FPR95 relative to MaxProb, PPL, and Entropy. Bold indicates the best value per model–dataset pair except CoE baselines. CoE-R and CoE-C results are taken from the original CoE paper (Wang et al., 2025b). Appendix B clarifies CoE usage (confidence estimation, not reranking) and motivates the token-centric KV-CoE design via layer-wise ablations. †These baseline results are reported on Llama3-8B-Instruct, while our experiments use the updated Llama3.1-8B-Instruct, so the numbers may not perfectly align.

We evaluate **KV-CoE** on two reasoning benchmarks from the original CoE work: MATH (Hendrycks et al., 2021) for multi-step arithmetic and TheoremQA (Chen et al., 2023) for theorem proving. Experiments are conducted on two popular instruction-tuned models: Llama-3.1-8B-Instruct (LlamaTeam, 2024) and Qwen2-7B-Instruct (QwenTeam, 2024).

We construct embeddings directly from the KV cache by extracting value vectors at every layer, concatenating across attention heads, and averaging over layers to obtain one embedding per token, all without storing hidden states. This reuse of cached information introduces negligible VRAM overhead and incurs minimal FLOPs consumption compared to vanilla CoE.

**Analysis.** As shown in Table 2, KV-CoE substantially outperforms baselines such as MaxProb, PPL, and Entropy on both MATH and TheoremQA. This demonstrates that the Chain-of-Embedding approach retains its strong discriminative power even when using KV cache-derived trajectories instead of hidden states. The token-level evolution captured by the KV cache provides a rich signal for identifying correct reasoning paths, especially in multi-step problems, all while adding negligible overhead since the cache is inherently available.

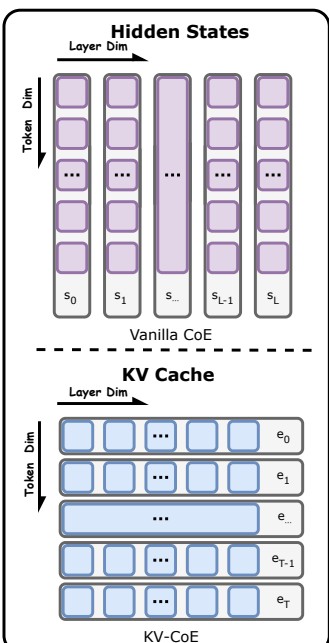

Figure 3: Vanilla CoE (top) aggregates hidden states across the token dimension to construct a representation for each layer, whereas **KV-CoE** (bottom) aggregates KV Cache across the layer dimension to construct a representation for each token.

## 6 FAST/SLOW THINKING WITH KV CACHE

Large Reasoning Models (LRMs) can operate in two modes: *fast thinking*, which produces short, direct answers, and *slow thinking*, which generates detailed, explicit step-by-step reasoning chains. (Yao et al., 2023; Lightman et al., 2024). Although slow thinking enhances reliability on complex tasks, it incurs substantial computational overhead by producing significantly more

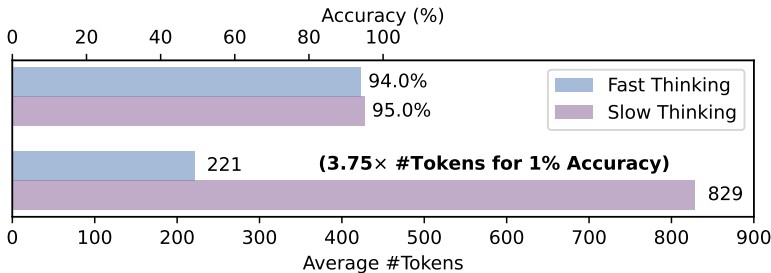

Figure 4: Comparison of efficiency and effectiveness of fast vs. slow thinking on GSM8K using Qwen3-32B. Slow thinking achieves slightly higher accuracy but at a much higher token cost.

tokens. For example, on GSM8K (Cobbe et al., 2021), Qwen3-32B (QwenTeam, 2025) slow thinking yields a marginal improvement in accuracy (0.95 vs. 0.94) while generating nearly four times the tokens, drastically increasing latency and cost as shown in Figure 4. This inefficiency motivates **adaptive reasoning**, where slow thinking is triggered selectively based on problem difficulty.

## 6.1 METHODOLOGY

We propose a method for adaptive reasoning that selects between fast and slow thinking on a per-instance basis to minimize unnecessary computation while maintaining accuracy. Our approach leverages the **KV cache** from the prompt encoding phase to make this decision, eliminating the need for additional forward passes.

**Key Idea.** Instead of predicting a binary "slow or fast" mode, we estimate a continuous difficulty score $d \in [0, 100]$ from the pooled KV cache representation:

$$d = f_\theta\left(\text{Pool}\left(KV_{1:T}^{(1:L)}\right)\right),$$

where $\text{Pool}(\cdot)$ aggregates keys and values across layers, heads, and token positions via mean pooling, and $f_\theta(\cdot)$ is a lightweight MLP classifier. This score determines whether to engage slow thinking.

**Switching Mechanism.** We control the reasoning mode by injecting special control tokens (<think> and </think>) during decoding:

- **Initial Decision:** Before generation starts, $d$ is compared to a predefined threshold $\tau$:
  - If $d > \tau$, prepend <think> to trigger slow thinking.
  - Otherwise, proceed with fast thinking.
- **Dynamic Adjustment During Decoding:** During generation, the difficulty score is re-computed from the updated KV cache at predefined checkpoints:
  - If $d < \tau_{\text{fast}}$ during slow thinking, append </think> to switch back to fast mode.
  - If $d > \tau_{\text{slow}}$ during fast mode, inject <think> to re-engage slow thinking and continue decoding with step-by-step reasoning.

This approach enables a fine-grained, difficulty-aware control over reasoning depth. Since the KV cache is already available from prompt encoding, both initial and ongoing difficulty assessments add negligible overhead. This significantly reduces token generation and latency for easy queries while allocating more resources to challenging problems. The overall workflow is illustrated in Figure 5.

**Training Data Construction.** To train the difficulty estimator $f_\theta(\cdot)$, we construct supervision signals from public reasoning datasets (training splits of GSM8K (Cobbe et al., 2021) and MATH (Hendrycks et al., 2021)). For each training question, we generate two candidate solutions using the base model: a *fast thinking* response (no chain-of-thought) and a *slow thinking* response (with chain-of-thought). We then extract the final answers and compare them against the ground truth.

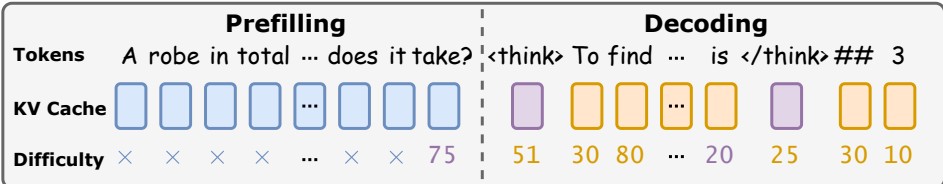

Figure 5: KVClassifier: special tokens are dynamically inserted to perform thinking-mode switching based on KV-derived difficulty scores.

Based on the outcomes, we assign a difficulty label $d \in \{0, 25, 75, 100\}$ reflecting the required reasoning depth:

- $d = 0$ (Very easy): Fast-thinking answer is correct and short ($< 128$ tokens).
- $d = 25$ (Moderate): Fast-thinking answer is correct but long ($\geq 128$ tokens).
- $d = 75$ (Hard): Fast-thinking answer is wrong, Slow-thinking answer is correct.
- $d = 100$ (Very hard): Both answers are incorrect.

This labeling scheme creates a natural difficulty progression, enabling $f_\theta(\cdot)$ to learn a smooth score that correlates with both correctness and reasoning effort. The token-length criterion distinguishes trivial questions from those needing lengthy outputs without explicit reasoning. The trained estimator provides the continuous difficulty score needed for our adaptive switching mechanism. See Appendix D for the exact label distribution and the prompt templates used to instantiate fast vs. slow thinking.

## 6.2 EXPERIMENTAL RESULTS

**Setup.** We evaluate our KV-cache-based fast/slow thinking mechanism on two reasoning benchmarks: GSM8K evaluation split (Cobbe et al., 2021) and MATH500 (OpenAI / HuggingFaceH4 / Vals AI, 2025). Our experiments compare two switching strategies:

- **One-step switch (KV-Classification):** This strategy makes a single decision at generation start based on the predicted difficulty score, committing to either slow or fast thinking for the entire decoding process. It functions as a *classification-style* controller.
- **Two-step switch (KV-Generative):** This method performs an initial mode selection and continuously monitors difficulty during decoding. If difficulty drops below $\tau_{\text{fast}}$ during slow thinking, it appends `</think>` to terminate reasoning early; if difficulty exceeds $\tau_{\text{slow}}$ during fast thinking, it injects `<think>` to engage slow thinking mid-generation. This implements a *generative-style* controller that dynamically adjusts reasoning depth.

We deploy both strategies on two open-weight models: DeepSeek-R1-14B (DeepSeek-AI, 2025) and Qwen3-8B (QwenTeam, 2025), evaluating their ability to selectively trigger slow thinking and reduce unnecessary token generation.

We construct representations by concatenating key and value tensors across all heads, summing over selected token positions, and averaging across selected layers without normalization, then feed the result into a two-layer MLP (hidden dimension 512, ReLU activation) for difficulty prediction. This design directly reuses the KV cache available during decoding, introduces negligible VRAM or FLOPs overhead, and functions as a modular component that can be seamlessly integrated into existing inference pipelines without modifications to the base model.

**Analysis.** As shown in Table 3, our KV-cache-based switching approach achieves an effective balance between accuracy and efficiency. For instance, on MATH500 using Qwen3-8B, two-step generative switching reduces average token count from 4,150 (full reasoning) to 727 ($5.7\times$ reduction) with only a minimal 3.2% accuracy drop. The one-step classification strategy is more conservative, using more tokens but achieving near-full-reasoning accuracy (0.604 vs. 0.610). Similar trends are observed on GSM8K, where KV-cache-based switching maintains high accuracy (up to 0.914)

| Dataset | Method | DeepSeek-R1-14B | Qwen3-8B |
|---------|--------|-----------------|----------|
| **GSM8K** | Fast Thinking | 0.845 / 218 | 0.904 / 211 |
| | Reasoning | 0.847 / 432 | 0.933 / 1632 |
| | KV-Classification | 0.845 / 218  -49.5% | 0.914 / 554  -66.1% |
| | KV-Generative | 0.835 / 242  -44.0% | 0.892 / 273  -83.3% |
| **MATH500** | Fast Thinking | 0.536 / 540 | 0.568 / 616 |
| | Reasoning | 0.590 / 1839 | 0.610 / 4150 |
| | KV-Classification | 0.578 / 1506  -18.1% | 0.604 / 3963  -4.5% |
| | KV-Generative | 0.566 / 657  -64.3% | 0.578 / 727  -82.5% |

Table 3: Comparison of accuracy and average token usage for fast thinking, full reasoning, and our KV-cache-based switching methods on GSM8K and MATH500. For each KV-based method, we report the result from the best hyper-parameter configuration identified in Appendix E.

while significantly reducing token consumption compared to full reasoning. These results demonstrate that difficulty scores derived from the KV cache generalize well across tasks and models, enabling efficient and effective adaptive reasoning with negligible overhead.

## 7 CONCLUSION

This work repurposes the KV cache, moving beyond its conventional role in decoding acceleration to unlock its potential as a versatile, cost-free representation. We demonstrate that although not designed as general-purpose embeddings, KV caches encode rich contextual information that can be effectively utilized for downstream tasks without incurring additional computational overhead. Our experiments establish two practical applications: (i) **Chain-of-Embedding**, where KV-derived embeddings match or surpass the performance of hidden-state embeddings, and (ii) **Fast/Slow Thinking Switching**, which uses KV-cache-based difficulty scores to enable adaptive reasoning-reducing token usage by up to $5.7\times$ with minimal accuracy loss. These findings position the KV cache as a deployment-friendly substrate for advanced inference techniques, opening new avenues for reusing inference-time artifacts to improve efficiency and controllability in LLMs.

## ACKNOWLEDGMENTS

The research work described in this paper was conducted in the JC STEM Lab of Machine Learning and Symbolic Reasoning funded by The Hong Kong Jockey Club Charities Trust.

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

## A    THE USE OF LARGE LANGUAGE MODELS (LLMs)

We acknowledge the use of a Large Language Model (LLM) to support the preparation of this manuscript. The LLM was employed exclusively for editorial purposes, such as refining the clarity of exposition, improving grammar and readability, and polishing the overall presentation. At times, it was also used to suggest alternative phrasings for technical descriptions in order to make the arguments more accessible.

Importantly, the LLM did not contribute to the conceptual development, methodology, or experimental design of this work. All ideas, including the proposal to treat the KV cache as a reusable representation, the development of KV-CoE for output-free self-evaluation, and the design of KV-based adaptive Fast/Slow Thinking Switching for token-efficient reasoning, were conceived and implemented solely by the authors. The LLM was not used to generate research results, proofs, or derivations.

All scientific claims, analyses, and conclusions presented in this paper remain the full responsibility of the authors. Any text initially produced with LLM assistance was carefully reviewed, revised, and verified before inclusion.

## B    CoE USAGE, KV-CoE DESIGN, AND TOKEN VS. LAYER DIMENSION

**CoE usage in reasoning benchmarks.**   In both the original CoE framework and our work, CoE is used as a *single-sample confidence estimator*. Given one completed reasoning trace, CoE outputs a scalar confidence score and is evaluated via AUROC/FPR95/AUPR. CoE is *not* used for pass@k reranking, and it does not modify decoding or generation.

**KV cache vs. hidden states across the layer dimension.**   The KV cache stores attention key/value tensors (per layer/head/token) and does not provide a drop-in substitute for full hidden states along the layer dimension. In particular, directly replacing hidden states with KV cache within the original layer-wise CoE design yields poor performance. This reflects a structural mismatch: the KV cache is inherently token-centric and head-structured, whereas hidden-state CoE explicitly tracks layer-wise representations.

**Ablation: layer-wise aggregation with KV-CoE.**   To make this limitation explicit, we evaluate a layer-wise variant of KV-CoE on `meta-llama/Llama-3.1-8B-Instruct`. Results are shown in Tables 4 and 5. Layer-wise KV-CoE performs substantially worse than our token-centric KV-CoE, confirming that KV cache should not be treated as a layer-wise embedding.

| Method | AUROC ↑ | FPR95 ↓ | AUPR ↑ |
|---|---|---|---|
| KV-CoE-R (layer) | 62.69 | 82.28 | 45.43 |
| KV-CoE-C (layer) | 64.09 | 81.16 | 47.78 |

Table 4:    Layer-wise    aggregation    ablation    of    KV-CoE    on    **MATH**    using `meta-llama/Llama-3.1-8B-Instruct`.

| Method | AUROC ↑ | FPR95 ↓ | AUPR ↑ |
|---|---|---|---|
| KV-CoE-R (layer) | 47.78 | 94.24 | 17.70 |
| KV-CoE-C (layer) | 48.61 | 92.68 | 18.02 |

Table 5:    Layer-wise    aggregation    ablation    of    KV-CoE    on    **TheoremQA**    using `meta-llama/Llama-3.1-8B-Instruct`.

**Design implication: token-centric KV-CoE.**   Motivated by the above, we redesign KV-CoE to operate along the *token dimension*, using token-wise aggregation that matches the semantics of the KV cache. Concretely, KV-CoE forms per-token embeddings from value vectors and aggregates

them over token positions (and optionally across heads/layers) to produce the trace-level confidence score.

**Pooling strategies summary.** For clarity, Table 6 summarizes the pooling/aggregation strategies used in this paper (see also L320–322 and L455–456).

| Task | Source | Head Agg. | Position Agg. | Layer Agg. |
|------|--------|-----------|---------------|------------|
| KV-CoE | Value vectors from KV cache | Concatenate | Per-token embedding | Average |
| Fast/Slow Thinking | Key + Value | Concatenate | Sum over selected tokens | Average (no normalization) |

Table 6: Summary of pooling strategies used in this work.

## C  HIDDEN-STATE AND CHANCE BASELINES ON MTEB CLASSIFICATION SUBSETS

To contextualize the MTEB classification results in Table 1, we report two additional baselines evaluated under the *same* downstream protocol (identical pooling, projection, and classifier setup as used for KV-cache embeddings).

**Hidden-state embeddings.** We extract hidden-state embeddings from `meta-llama/Llama-3.1-8B-Instruct` and apply the same aggregation pipeline used for KV-cache embeddings. Table 7 shows that hidden-state embeddings achieve accuracies close to those of KV-cache embeddings under identical pooling and projection, suggesting that the observed gap to dedicated embedding models is dominated by the downstream protocol rather than the choice between hidden states vs. KV caches.

| Dataset | Accuracy (Hidden State) |
|---------|------------------------|
| AmazonCounterfactual | 0.3530 |
| DBpedia | 0.5937 |
| FinancialPhrasebank | 0.6254 |
| TweetTopic | 0.3714 |

Table 7: Accuracy of hidden-state embeddings on selected MTEB classification tasks under the same pooling/projection pipeline as KV-cache embeddings.

**Random-embedding baseline.** We additionally include a random-embedding baseline to sanity-check that the evaluation pipeline is not degenerate. Concretely, we replace the model-derived embedding with an i.i.d. random vector $e_{\text{rand}} \in \mathbb{R}^d$ (matched to the embedding dimension $d$ used in our pipeline), while keeping the *same* pooling/projection interface (where applicable) and the same downstream classifier and evaluation protocol. Table 8 reports the resulting accuracy. The consistent margin between random embeddings and hidden/KV embeddings—particularly on DBpedia and FinancialPhrasebank—indicates that the pipeline is functional and that model-derived embeddings capture non-trivial task structure beyond what is achievable with unstructured random features.

| Dataset | Accuracy (Random) |
|---------|-------------------|
| AmazonCounterfactualClassification | 0.5224 |
| DBpediaClassification | 0.0716 |
| FinancialPhrasebankClassification | 0.3274 |
| TweetTopicClassification | 0.1591 |

Table 8: MTEB classification performance when representations are replaced by a random baseline embedding $e \in \mathbb{R}^{256}$, whose entries are drawn i.i.d. from a standard normal distribution.

**Scope.** These MTEB subsets are reported only to illustrate that KV-derived embeddings are *cost-efficiently comparable* under our protocol; we do not claim they are superior to dedicated embedding models, nor that they are reliable for global semantic retrieval across diverse domains.

## D  DIFFICULTY LABELS AND PROMPT CONTROL FOR FAST/SLOW THINKING SWITCHING

**Difficulty labels for training the switching model.** We assign each instance a discrete difficulty label based on the correctness of *fast* vs. *slow* thinking outputs, using the following four-level scheme:

| Difficulty | Description | Count |
|---|---|---|
| 0 | Both fast and slow thinking are correct (very easy) | 286 |
| 25 | Both are correct but require longer generation | 3,467 |
| 75 | Only slow thinking is correct | 860 |
| 100 | Neither is correct | 2,887 |
| Total | | 7,500 |

Table 9: Difficulty labels used to train the fast/slow switching estimator.

**Fast vs. slow thinking prompts.** We use a Large Reasoning Model (LRM) that supports explicit reasoning via `<think>` blocks. Fast/slow thinking is controlled *purely by prompt*, not by KV cache. Specifically, after the user prompt: (i) **Fast thinking** inserts an empty thinking block `<think>\n</think>`; (ii) **Slow thinking** inserts an open thinking block `<think>\n`, which encourages step-by-step reasoning. Model-specific special tokens (e.g., `<|im_start|>`, `<|im_end|>`) are adapted per backbone; the KV-based controller only chooses between the fast and slow prompt variants.

## E  HYPER-PARAMETER SELECTION FOR KVCLASSIFIER

To better understand the effect of hyperparameters on KV-based classification, we conduct a systematic study by varying the number of layers pooled and the number of tokens selected from the end of the sequence. Importantly, we fix the *total amount of KV data* to be approximately constant across configurations (256 token × layer units). This ensures a fair comparison: for example, selecting 8 layers with 32 tokens, 4 layers with 64 tokens, or 2 layers with 128 tokens, all yield the same KV budget.

| Model | Dataset | Method | 8L, Len=32 | 4L, Len=64 | 2L, Len=128 |
|---|---|---|---|---|---|
| DeepSeek-14B | GSM8K | KV-Classification | 0.845 / 218 | 0.845 / 218 | **0.845** / 218 |
| | | KV-Generative | **0.835** / 242 | 0.825 / 232 | 0.805 / 217 |
| | MATH500 | KV-Classification | 0.536 / 540 | 0.550 / 905 | **0.578** / 1506 |
| | | KV-Generative | 0.538 / 524 | 0.550 / 544 | **0.566** / 657 |
| Qwen3-8B | GSM8K | KV-Classification | 0.904 / 211 | 0.904 / 217 | **0.914** / 554 |
| | | KV-Generative | **0.892** / 273 | 0.886 / 276 | 0.881 / 257 |
| | MATH500 | KV-Classification | 0.570 / 736 | 0.598 / 3673 | **0.604** / 3963 |
| | | KV-Generative | **0.578** / 727 | 0.524 / 933 | 0.550 / 837 |

Table 10: Hyper-parameter selection results for KV-Classification and KV-Generative. Values are reported as Accuracy / #Tokens. Best accuracy for each dataset–method pair is in bold.
Table 10 summarizes the results on GSM8K and MATH500 for both DeepSeek-R1-14B and Qwen3-8B, under KV-Classification and KV-Generative settings. We observe that while performance varies slightly with the allocation of layer vs. token depth, the overall trends are consistent: (i) accuracy remains competitive across different allocations, and (ii) increasing token coverage (e.g., 2L × 128) tends to favor more complex datasets such as MATH500, whereas shallow but wider layer coverage (e.g., 8L × 32) can suffice for GSM8K.

