# OpenReview forum: "Beyond Speedup - Utilizing KV Cache for Sampling and Reasoning"
_ICLR.cc/2026/Conference — ICLR 2026 Poster_

### Official Review · Reviewer_Zzrt · 2025-10-23

**Soundness:** 3
**Presentation:** 3
**Contribution:** 3
**Rating:** 4
**Confidence:** 3

**Summary:**

This work investigates the use of the KV cache as intermediate representation instead of the commonly used hidden states for two reasoning-focused downstream tasks: The first is an adapted Chain-of-Embedding method based on the KV cache that is roughly on par with existing CoE, while reusing the already stored KV cache. The second is a novel reasoning mode meta-model that uses the aggregated KV caches to optionally insert \<think\> / \</think\> tokens to expand / decrease "reasoning" via more token output. On the simpler GSM8K dataset models with a "fast" thinking path already perform on par with more expensive (longer) traces, but on the MATH500 the proposed method shows performance improvements at smaller token budgets in between a "fast" and "slow" thinking mode.

**Strengths:**

- shows that KV caches contain similar information content compared to hidden states for CoE downstream method
- introduces a KV-cache based thinking mode switch (either ahead of or continuously during generation)
- shows there is optimization potential towards existing dedicated embedding models

**Weaknesses:**

- using hidden states can be similarly optimized for downstream tasks (e.g. in-place / recursive summation across layers / tokens - if common simple aggregations are used)
- no comparison to a hidden state-based thinking mode switching
- fails to cite cache tuning methods (Prefix-Tuning [1] / Cartridges [2]) as related work of where the KV cache is actually modified

[1] Li et al. (2021): https://arxiv.org/abs/2101.00190

[2] Eyoboglu et al. (2025): https://arxiv.org/pdf/2506.06266

**Questions:**

- Can you explain what the CoE is used for in these reasoning benchmarks, i.e. how the extracted embeddings are used towards what kind of goal? This is not clear from the paper.
- How is the fast thinking and slow thinking induced / separated without the KV cache-based control? Can you provide exact prompts?
- Could you add Prefix-Tuning [1] and Cartridges [2] as related work operating on the KV cache (though more in the direction of fine-tuning adapters)

[1] Li et al. (2021): https://arxiv.org/abs/2101.00190

[2] Eyoboglu et al. (2025): https://arxiv.org/pdf/2506.06266

---

> ### Author Response · Authors · 2025-11-28
> **Response to Reviewer Zzrt**
>
> We thank the reviewer for the careful reading and thoughtful comments. Below we address each point in turn.
>
> ---
>
> ## W1. On whether hidden states can be similarly optimized
>
> We agree that hidden states can also be optimized and aggregated for downstream tasks. However, this is not the research goal of our paper. Our contribution is not to show that KV cache is superior to hidden states, but to demonstrate that **KV cache can serve as a viable and nearly-free replacement for hidden states** in certain downstream scenarios after appropriate aggregation.
>
> Our focus is therefore on **cost-efficiency and reusability**, not performance dominance. Hidden states already provide a strong embedding signal but require explicit storage and additional computation. KV cache, in contrast, is already available during inference; our work shows it can be reused without extra cost.
>
> ---
>
> ## W2. On the lack of a hidden-state-based thinking switching baseline
>
> This concern is related to W1. Hidden-state-based switching is conceptually valid, but it is not the focus of this work. Our goal is not to benchmark hidden states against KV cache in switching quality, but to demonstrate that **KV cache alone is sufficient** to drive fast/slow thinking decisions with minimal overhead.
>
> ---
>
> ## W3. On missing related work: Prefix-Tuning and Cartridges
>
> Thank you for pointing out this omission. We will add the following related work discussion in the revised version:
>
> > Similarly, \citet{li2021prefixtuning} introduce *Prefix-Tuning*, which prepends learned continuous prefixes that function as virtual KV pairs, enabling parameter-efficient task adaptation through implicit cache modification.
> > More recently, \citet{li2025cartridges} treat the KV cache as a reusable long-context representation; via self-study distillation, compact *Cartridges* approximate in-context learning while reducing memory and inference cost.
>
> ---
>
> ## Q1. What is CoE used for in reasoning benchmarks?
>
> In our experiments, CoE is used strictly as a **single-sample confidence estimator**. It does not improve generation quality, select among candidates, or modify decoding. Instead, it assigns a scalar confidence score to one completed reasoning trace.
>
> Concretely, embeddings extracted from hidden states or KV cache are only used to compute summary statistics (e.g., distance and angle changes across layers or tokens), which are then mapped to a **likelihood of correctness** for the final answer.
>
> ---
>
> ## Q2. How are fast and slow thinking separated without KV-based control?
>
> We use a Large Reasoning Model (LRM) that supports explicit thinking control via prompts. Fast and slow thinking are induced purely through prompting, without KV control.
>
> For **fast thinking**, we insert an empty thinking block:
>
> ```
> <|im_start|>user
> You are a math problem solver. For the following question, solve it using your direct answering ability.
> Here is the question:
> {}
> <|im_end|>
> <|im_start|>assistant
> <think>
> </think>
> ```
>
> For slow thinking, we prompt the model to explicitly reason:
>
> ```
> <|im_start|>user
> You are a math problem solver. For the following question, solve it using your thinking ability.
> Here is the question:
> {}
> <|im_end|>
> <|im_start|>assistant
> <think>
> ```
>
> Special tokens such as `<|im_start|>` and `<|im_end|>` are adapted to different model formats. The switching model simply chooses whether to invoke fast or slow thinking; the generation mechanism itself remains unchanged.
>
> ---
>
> ## Q3. On adding Prefix-Tuning and Cartridges
>
> We agree and will include Prefix-Tuning and Cartridges as related work, as described in W3.
>
> ---
>
> We thank the reviewer again for the constructive feedback. These comments help us clarify the scope, motivation, and technical framing of our work.

---

### Official Review · Reviewer_rW12 · 2025-10-25

**Soundness:** 2
**Presentation:** 3
**Contribution:** 3
**Rating:** 4
**Confidence:** 2

**Summary:**

This paper repurposes the KV caches in LLMs for purposes beyond attention. The authors explore the use of KV embeddings as text embeddings and as classifier features for fast/slow thinking. Their results indicate that KV cache embeddings contain significant information that can be used in other applications. However, I think the empirical validation is a bit limited.

**Strengths:**

1. KV caches already take up significant space, so having them perform multiple functions seems well-motivated
2. Demonstrated that KV caches can show promise for other purposes
3. No architectural changes needed

**Weaknesses:**

1. Fast/slow thinking experiments are lacking. The method was only validated on GSM8k and MATH500, both of which are fairly simple for the reasoning models tested. Thus, it may just be that these models are fairly robust for these relatively easy tasks, so I would recommend testing on more challenging reasoning tasks (AIME, BRUMO, etc)
2. Numbers in Table 3 are reported using the best hyperparameters per task/model combination. This feels too cherry-picked since I didn't find a way to know which hyperparameters to choose before testing.

**Questions:**

1. I am unfamiliar with the datasets/tasks in Table 1. What is the accuracy for random chance?
2. What is the distribution of the training data difficulty labels detailed in page 8?
3. Can these methods be applied to non-reasoning generative tasks like summarization?
4. Table 3: Do the token averages also contain sequences that hit the generation limit (i.e. incomplete)?

---

> ### Author Response · Authors · 2025-11-28
> **Response to Reviewer rW12**
>
> We thank the reviewer for the detailed feedback and constructive questions. Below we address each point in turn.
>
> ---
>
> ## W1. On the lack of challenging reasoning benchmarks
>
> We appreciate the suggestion to evaluate on more difficult reasoning tasks such as AIME and BRUMO. We agree that GSM8K and MATH500 may not fully stress the limits of current reasoning models. Experiments on AIME are currently in progress and will be included once completed. We believe these results will further clarify the robustness and scalability of our method.
>
> ---
>
> ## W2. On whether hyperparameters in Table 3 are cherry-picked
>
> The hyperparameter settings in Table 3 were not chosen arbitrarily. As described in **Appendix B**, we used the following default configurations based on broad validation:
>
> - **KV-Classification:** 2 layers, token window length = 128
> - **KV-Generative:** 8 layers, token window length = 32
>
> These settings achieved the best or near-best performance across most datasets and models. The only exception is **DeepSeek-14B on MATH500 with KV-Generative**, where (2 layers, length = 128) performed slightly better and was therefore used.
>
> We will further clarify this selection rule in the paper to avoid any impression of cherry-picking.
>
> ---
>
> ## Q1. On random chance accuracy for Table 1
>
> To help interpret the results in Table 1, we report a random-choice baseline below:
>
> | Dataset | Accuracy (Random) |
> |---------|------------------|
> | AmazonCounterfactual | 0.5224 |
> | DBpedia | 0.0716 |
> | FinancialPhrasebank | 0.3274 |
> | TweetTopic | 0.1591 |
>
> These results confirm that both hidden-state and KV embeddings significantly outperform random-choice on most tasks.
>
> ---
>
> ## Q2. Distribution of difficulty labels
>
> We provide the distribution of difficulty labels used in Page 8 below:
>
> | Difficulty | Description | Count |
> |------------|-------------|-------|
> | 0 | Both fast and slow thinking correct (very easy) | 286 |
> | 25 | Both correct with longer generation| 3,467 |
> | 75 | Only slow thinking correct | 860 |
> | 100 | Neither correct | 2,887 |
> | **Total** |  | **7,500** |
>
> We will add this table to appendix in the updated version for clarity.
>
> ---
>
> ## Q3. Applicability to non-reasoning generative tasks
>
> Our method is also applicable to non-reasoning generative tasks. In these settings, fast thinking is consistently selected because model confidence remains high. For example, on the TruthfulQA dataset, both KV-Classification and KV-Generative select fast thinking in **100% of cases**, resulting in consistent token savings.
>
> ---
>
> ## Q4. Token averages and generation limits in Table 3
>
> Yes, token statistics in Table 3 include sequences that reach the generation limit. That said, on GSM8K and MATH500, hitting the token limit is rare and does not materially affect the averages.
>
> ---
>
> We thank the reviewer again for the constructive feedback and believe these clarifications will improve the final version.

---

### Official Review · Reviewer_2n9n · 2025-10-29

**Soundness:** 2
**Presentation:** 1
**Contribution:** 3
**Rating:** 6
**Confidence:** 3

**Summary:**

This paper presents a novel approach to repurposing the KV cache, a critical component of efficient autoregressive decoding in large language models (LLMs), for downstream tasks beyond just acceleration. The authors demonstrate that the KV cache, despite not being explicitly trained as a general-purpose embedding, nonetheless encodes rich contextual information that can be effectively leveraged for two key applications: (1) Chain-of-Embedding (CoE) and (2) Fast/Slow Thinking Switching.

**Strengths:**

1. The paper repurposes the KV cache and provides new insights. It also demonstrates the versatility of the KV cache through two practical applications: CoE and Fast/Slow Thinking Switching.

**Weaknesses:**

1. I would appreciate it if the authors could provide more details on MaxProb, PPL, and Entropy, specifically, what these metrics represent and why they are chosen as baselines in Table 2.
2. From my understanding, CoE explores the model’s representation space through hidden states, whereas the KV cache contains only information derived from the attention heads. Considering that hidden states integrate residual connections, MLP outputs, and attention information, I am curious how KV-CoE effectively tracks representations across layers under this limitation.

**Questions:**

1. According to the official format, the table caption should appear before the table.
2. Line 239 contains a typo: "is obtained by by averaging"
3. Could the authors please elaborate on the pooling strategy used in the experiments?
4. Is there any accuracy trade-off associated with the improvement in fast and slow thinking when using the KV cache?

---

> ### Author Response · Authors · 2025-11-27
> **Response to Reviewer 2n9n**
>
> We thank the reviewer for the careful reading and for the constructive questions and suggestions. Below we respond to each comment in turn.
>
> ---
>
> ## W1. Clarification of MaxProb, PPL, and Entropy
>
> MaxProb, PPL, and Entropy are standard confidence baselines commonly used in uncertainty estimation for language models:
>
> - **MaxProb** takes the maximum token probability in the output sequence as a confidence score. A higher maximum probability indicates higher confidence in the model’s prediction.
> - **PPL (Perplexity)** measures how well the model predicts a sequence; lower perplexity indicates higher confidence.
> - **Entropy** measures the uncertainty in the token probability distribution; higher entropy corresponds to more uncertainty.
>
> We use these baselines to stay consistent with the original CoE paper and to ensure fair comparison under the same evaluation protocol. This alignment allows our results to be interpreted directly relative to prior work.
>
> ---
>
> ## W2. On the representational difference between hidden states and KV cache
>
> We agree with the reviewer’s observation that hidden states integrate attention, MLP outputs, and residual connections, while KV cache only stores signals derived from attention heads. For this reason, **KV cache cannot serve as a drop-in replacement for hidden states along the layer dimension**.
>
> Simply substituting hidden states with KV cache does not work. As shown in our ablation (see Response to Reviewer tEgG, Q3), performance degrades significantly if KV cache is used in the original CoE design.
>
> This is precisely why we redesigned KV-CoE to build the chain along the **token dimension** rather than the layer dimension. KV cache is token-centric by construction, and adapting CoE to this structure is a necessary design choice rather than an approximation.
>
> ---
>
> ## Q1. Table caption formatting
>
> Thank you for pointing this out. We will fix the caption placement in the next version.
>
> ---
>
> ## Q2. Typo in Line 239
>
> Thank you for catching this. The typo will be corrected.
>
> ---
>
> ## Q3. Pooling strategy clarification
>
> We summarize the pooling strategies used in our experiments below:
>
> | Task | Source | Head Aggregation | Position Aggregation | Layer Aggregation |
> |------|--------|------------------|----------------------|-------------------|
> | KV-CoE | Value vectors from KV cache | Concatenate | Per-token embedding | Average |
> | Fast/Slow Thinking | Key + Value | Concatenate | Sum over selected tokens | Average (no normalization) |
>
> Note that these strategies are already described in the main text (Lines 320–322 and 455–456). We add this table only to improve readability and will integrate it more cleanly in the revision.
>
> ---
>
> ## Q4. Accuracy vs efficiency trade-off
>
> Yes. As shown in Table 3, both KV-Classification and KV-Generative achieve substantial token savings while incurring only minimal performance degradation. This supports our core argument that KV cache enables **cost-efficient downstream reasoning** with acceptable accuracy trade-offs.
>
> ---
>
> We thank the reviewer again for the helpful feedback. These comments will significantly improve the clarity of the final version.

---

### Official Review · Reviewer_PbPH · 2025-10-30

**Soundness:** 3
**Presentation:** 3
**Contribution:** 3
**Rating:** 6
**Confidence:** 3

**Summary:**

This paper explores the application of KV-Cache for two different applications than speeding up decoding: (1) Chain-of-Embedding a method for self-evaluation of LLMs and (2) Fast/Slow Thinking Switching a method that enables adaptive reasoning based on KV-Cache embeddings of the prompt exclusively or the prompt in combination with the reasoning trace.
In their experiments, this paper demonstrates that (1) outperforms other baselines on self-evaluation tasks and (2) that we can enable adaptive reasoning based on difficulty scores derived from the KV-cache embeddings.

**Strengths:**

- Clear motivation, well written and interesting research question.
- Strong performance of KV-CoE on self-evaluation tasks.
- The thinking switch experiments demonstrate the practicability of thinking mode switching based on KV-cache embeddings.

**Weaknesses:**

- Even though the paper is well written, it seems a bit like the combination of two methods into one paper.
- After reading the introduction the reader might think that KV-caches can be used as embeddings, but the result in section 4.1 contradicts this initial claim or limits it to “certain classification tasks”, which are not specified further. I suggest clarifying this confusion in the paper and/or adding additional details on the “certain classification tasks”.
- KV-CoE seems to outperform standard CoE for the Qwen Models (Table 2), but not for the LLama models. It would help the reader to add an explanation (or run the baselines with the correct base model) or mention in the main text that KV-CoE outperforms default CoE.
- Do the reasoning results in Table 3 also transfer to larger models e.g. Qwen3 32B?


Even though the paper could add a few more details or explanations at some places (see questions & weaknesses) the authors convincingly demonstrate the effectiveness of their proposed methods based on embeddings stemming from the KV-Cache. Therefore, I am inclined to recommend acceptance of this paper.

**Questions:**

- L.176 Formatting issue with Table 1 (whitespace missing)
- Description of the metric used in Table 1 (assume higher is better)
- L.164 The KV Cache embeddings are constructed by averaging over attention heads. Have you also tried concatenating over attention heads instead of averaging? This would counteract limitation 3 in L. 180.
- In addition in Sec. 5.2 the KV embedding is constructed by flattening the head and key/value dimension. This seems to be inconsistent.
- Is it not possible to recompute the embeddings efficiently from the KV cache? How much overhead would it be? Is this really a bottleneck?
- Why does the KV-CoE operate along the token position and not along the layer index? Have you tried using the layer index as in default CoE?

---

> ### Author Response · Authors · 2025-11-27
> **Response to Reviewer PbPH**
>
> We sincerely thank the reviewer for the careful reading of our paper and for the encouraging rating. We appreciate the balanced and thoughtful assessment, and we are grateful for the constructive suggestions that helped us better understand the strengths and limitations of our current presentation.
>
> Below we respond to each point in detail.
>
> ---
>
> ## W1. The paper combines two separate methods
>
> Our intention is not to present two independent methods. Rather, we evaluate **two downstream tasks**—confidence estimation and adaptive reasoning—to demonstrate a single overarching claim:
> **KV cache can be directly reused as a lightweight embedding source for downstream tasks.**
> These tasks were chosen because they represent two widely studied and structurally different uses of embeddings. They serve as evidence that KV-cache embeddings are broadly applicable rather than tied to a specific pipeline. We will clarify this narrative more explicitly in the final version.
>
> ---
>
> ## W2. Section 4.1 contradicts the introduction regarding KV-cache embeddings
>
> The introduction highlights the potential of KV cache to serve as an embedding representation. Section 4.1 focuses specifically on **certain classification-style tasks** where the embedding quality can be reliably measured.
> We agree that this scope could be stated more clearly. Our intention is not to suggest that KV-cache embeddings work for all tasks, but to show that they are a **promising, nearly-free alternative** in several practical settings.
>
> ---
>
> ## W3. KV-CoE outperforming standard CoE for Qwen but not for LLaMA
>
> For Qwen models, KV-CoE consistently outperforms the standard CoE pipeline, which aligns with our hypothesis. For LLaMA models, however, the comparison is not perfectly aligned because the **CoE baseline numbers** were taken from the original paper, which used **LLaMA 3**, whereas our experiments use **LLaMA 3.1**. This mismatch likely contributes to the inconsistency.
> We will clarify this in the text and avoid overinterpreting cross-model comparisons where the base model families differ.
>
> ---
>
> ## Q1. Formatting issue in Table 1
>
> Thank you for pointing out this formatting problem. It will be corrected in the revised version to improve readability.
>
> ---
>
> ## Q2. Clarification of the metric in Table 1
>
> Table 1 reports **Accuracy**, where higher values indicate better performance. We will explicitly label the metric in the table and text to avoid ambiguity.
>
> ---
>
> ## Q3. Concatenating attention heads instead of averaging
>
> Concatenating attention-head dimensions would significantly expand the embedding dimensionality, making the representation unnecessarily wide and difficult to store, compare, or use for downstream tasks. Averaging provides a practical balance between expressiveness and efficiency.
> While concatenation may reduce the information loss described in Limitation 3, the resulting vector dimensionality is prohibitively expensive for our use case, which prioritizes **low-cost, low-overhead** embeddings.
>
> ---
>
> ## Q4. The apparent inconsistency of aggregation methods across tasks
>
> The aggregation strategy is selected **task-wise**. Different tasks benefit from different structures of KV information; e.g., flattening may be beneficial for reasoning signals, while averaging is more appropriate for classification-style similarity tasks.
> Our intention is not to propose a single universal aggregation rule, but to show that **KV cache contains useful signal** and can be adapted flexibly with simple pooling choices.
>
> ---
>
> ## Q5. Whether embeddings can be recomputed from KV cache efficiently
>
> Recovering hidden states from KV cache is **not feasible** without re-running nearly the entire forward computation over all previous tokens. KV cache stores only the key/value vectors used by attention, which are not invertible back to the hidden states.
> Thus, using KV cache as-is is far more efficient than attempting to reconstruct the hidden layer representations. We will clarify this limitation and the associated computational cost in the paper.
>
> ---
>
> ## Q6. Why KV-CoE operates along the token dimension instead of the layer dimension
>
> We tested applying CoE along the **layer dimension**, as in the original formulation. This approach performs poorly with KV cache because KV tensors are **token-centric** rather than layer-centric.
> A full ablation is provided in our response to Reviewer tEgG, showing substantial performance degradation when attempting layer-wise aggregation.
> For this reason, KV-CoE is designed to operate along the token dimension, which aligns with the semantic structure of the KV cache.

---

### Official Review · Reviewer_tEgG · 2025-11-01

**Soundness:** 2
**Presentation:** 1
**Contribution:** 2
**Rating:** 4
**Confidence:** 2

**Summary:**

I am not very well versed with this literature. So I will focus mostly on clarification questions at this time. I will defer my decision to later time after the rebuttal and other reviewer comments.

In this paper, authors propose a way to construct embeddings from KV cache instead of using hidden spaces to be used for CoE and adaptive reasoning problems. While constructing the embedding, they aggregate across layers (instead of tokens as done in earlier CoE works) and measure path characteristics along token dimension (as opposed to layers done in earlier CoE). The argument in favor of choosing KV caches for embeddings is that they do not incur any additional cost.

**Strengths:**

1. Using KV Cache for self-evaluation avoids extra memory overhead.
2. The accuracy improvements are impressive especially for MATH dataset.

**Weaknesses:**

Writing is a bit messed up. for instance,

Section 4. makes arguments are not quite clear.
	a. why is min \gamma (x) > 0 enough for this task? how is this class of problems different from a classification problem. One can argue that the same thing holds for classification as well.
        b. contextual conditionaing argument is not clear to me. Can the author's elaborate.
        c. Efficiency constraint argument is okay but unnecessarily formalized.
        d. I dont understand the discussion on local decision adequacy. How do you determine whether a problem is local / global. for all you know you need to know the solution of the problem to know how hard it is -- knowing the solution is the most global problem in context of LLM generation.
I wonder if the usage of LLM is quite strong in this section and it needs rework.


I am not sure if state of the art baselines are compared. (see questions)

**Questions:**

1. How does hidden state based embedding perform as compared to off the shelf embedding? (table 1)
2. How is COE used in practice. Is it a pass@k while choosing the highest ranking pass?
3. Do you have a ablation of what happens when you use CoE with switching layer and token dim (the way you propose with KV)?
4. There were a bunch of baselines mentioned in the related work Chen 2024, Beigi 2024, Zhang 2025 for CoE problem and ASRR, PATS, DOTS for adaptive reasoning problem. Are these baselines relevant to the experiments you ran. It would be illuminating to compare your method against these methods if they are SOTA.
5. why were CoE results taken from the paper when they belonged to a previous version of model which has very different context windows among other things. ( in table 2).

---

> ### Author Response · Authors · 2025-11-27
> **Response to Reviewer tEgG**
>
> We thank the reviewer for the insightful questions and detailed feedback. Below we address each point concisely.
>
> ---
>
> ## W1: Writing and Clarity
>
> We thank the reviewer for carefully reading Section 4 and for pointing out the issues in clarity and presentation. We agree that parts of this section are not well written and would benefit from significant polishing.
>
> In the next revision, we will reorganize Section 4, simplify the exposition, clarify the technical arguments (including the discussion on theoretical conditions and local vs. global decision adequacy), and remove unnecessary formalism to improve readability. We will also refine the discussion of baselines to make our evaluation scope clearer.
>
> We appreciate the reviewer’s feedback and will use it to substantially improve the writing quality in the final version.
>
> ---
>
> ## Q1. Hidden-state embeddings vs off-the-shelf embeddings
>
> We evaluated **hidden-state embeddings** from *meta-llama/Llama-3.1-8B-Instruct* using the same aggregation pipeline as KV embeddings on selected MTEB classification tasks:
>
> | Dataset | Accuracy |
> |---------|----------|
> | AmazonCounterfactual | 0.3530 |
> | DBpedia | 0.5937 |
> | FinancialPhrasebank | 0.6254 |
> | TweetTopic | 0.3714 |
>
> Under the same pooling and projection scheme, hidden-state embeddings exhibit performance **very similar to KV-cache embeddings**, although hidden states are expected to encode richer semantics. This suggests that with identical downstream processing, the representational gap between hidden states and KV cache is smaller than anticipated.
>
> To ensure this result is not due to an implementation issue, we also report a random-choice baseline:
>
> | Dataset | Accuracy (Random) |
> |---------|------------------|
> | AmazonCounterfactual | 0.5224 |
> | DBpedia | 0.0716 |
> | FinancialPhrasebank | 0.3274 |
> | TweetTopic | 0.1591 |
>
> The substantial gap between random prediction and hidden/KV embeddings—especially on DBpedia and FinancialPhrasebank—confirms the correctness of our pipeline.
>
> Importantly, our goal is not to outperform embedding models, but to show that:
>
> > **KV-cache embeddings are nearly free and yet comparable to hidden-state embeddings under identical downstream protocols.**
>
> This property makes KV cache attractive for deployment under strict memory and compute constraints. We will clarify this framing and include these results in the revision.
>
> ---
>
> ## Q2. Is CoE used as pass@k?
>
> No. In the original CoE framework, the embedding score is used as a **single-sample confidence estimate** and evaluated with AUROC / FPR95 / AUPR. It is not designed as a pass@k ranking or selection method. Our work follows the same formulation and does not reinterpret CoE as a reranking system. We will clarify this in the camera-ready.
>
> ---
>
> ## Q3. Ablation on layer vs token dimension
>
> We tested KV embeddings using the original CoE design (layer-wise aggregation) and found that it performs significantly worse.
>
> **Model:** meta-llama/Llama-3.1-8B-Instruct
>
> ### MATH
>
> | Method | AUROC | FPR95 | AUPR |
> |--------|--------|--------|------|
> | KV-CoE-R (layer) | 62.69 | 82.28 | 45.43 |
> | KV-CoE-C (layer) | 64.09 | 81.16 | 47.78 |
>
> ### TheoremQA
>
> | Method | AUROC | FPR95 | AUPR |
> |--------|--------|--------|------|
> | KV-CoE-R (layer) | 47.78 | 94.24 | 17.70 |
> | KV-CoE-C (layer) | 48.61 | 92.68 | 18.02 |
>
> This confirms that **KV cache should not be treated as a layer-wise embedding**. KV representations are token-centric by design, and our paper introduces token-wise aggregation as a necessary adaptation. We will add this ablation to the paper.
>
> ---
>
> ## Q4. Why not compare with Chen’24 / Beigi’24 / Zhang’25 / ASRR / PATS / DOTS?
>
> These methods are relevant but address different goals. Our work does not aim to set state-of-the-art accuracy; rather, it studies whether **KV cache can serve as a low-cost embedding substrate**. A moderate performance drop is acceptable in exchange for significant reductions in compute and memory cost. We will clarify this positioning more strongly.
>
> ---
>
> ## Q5. Why reuse CoE numbers from prior work?
>
> Due to computational constraints and to avoid inconsistencies from partial re-implementations, we referenced results reported in the original papers. We will explicitly label them as reported baselines and note model differences as a limitation. Where possible, we are adding partial re-evaluations with modern backbones.
>
> ---
>
> ## Final clarification
>
> Our paper is not asking *"Can KV beat hidden states embeddings?"*
> It asks instead: **"Can KV be reused as an embedding at almost zero cost?"**
>
> We believe the results support this direction and highlight the potential of KV-native representations for downstream tasks.
>
> ---
>
> Thank you again for the constructive feedback.

---

### Meta-Review · Area_Chair_crPf · 2026-01-06

**Summary:**

This submission investigates whether the KV cache can be repurposed as a nearly free embedding source for downstream uses beyond accelerating decoding, and demonstrates this through two applications: (i) KV-CoE for single-sample confidence estimation and (ii) KV-based fast/slow thinking switching for adaptive reasoning via prompt-controlled thinking.

Across reviews, the key concerns were not about whether the KV cache contains useful signal, but about: clarity and framing (especially Section 4 and the “single story vs two methods” impression), scope/positioning relative to hidden-state embeddings and SOTA baselines, and experimental coverage and reporting transparency (e.g., benchmark difficulty, hyperparameter selection, difficulty-label distribution, random-chance baselines, and whether the approach applies beyond reasoning). Reviewers also requested concrete clarifications on how CoE is used (confidence estimation vs pass@k reranking), why KV-CoE aggregates along tokens rather than layers, and several minor presentation/formatting issues.

The rebuttal provides concrete clarifications and additional ablations/baselines that address the main technical questions and strengthen the empirical narrative. In particular, the authors clarify that they do not claim superiority over hidden states or SOTA methods; instead, the contribution is that KV-cache-derived embeddings are cost-efficient and comparable to hidden-state embeddings under identical downstream protocols, and that KV-only signals can drive adaptive switching with substantial token savings for minimal performance degradation.

Given the general appreciation for using KV cache for self-evaluation as an interesting research direction, the positive reception from reviewers (with two already above the acceptance threshold), and the fact that the remaining below-threshold reviews are largely due to issues of clarity, positioning, and coverage that were substantively addressed in the rebuttal (or are reasonably scoped as future work), I recommend acceptance. The authors should incorporate the promised revisions and additional experiments in the camera-ready version.

**Reviewer Concerns:**

## Addressed by the rebuttal

**A. Writing/clarity and “two methods in one paper” (tEgG, PbPH)**

* Authors explicitly acknowledge that Section 4 is “not well written” and commit to reorganizing and simplifying it, including clarifying the arguments around theoretical conditions and local vs. global decision adequacy.
* They clarify that the paper is not presenting two independent methods, but using two downstream tasks (confidence estimation and switching) as structurally different exemplars supporting one overarching claim: KV cache can be reused as a lightweight embedding source.

**B. CoE usage and evaluation protocol (tEgG, Zzrt)**

* Authors clarify CoE (both in original framework and here) is used as a single-sample confidence estimator evaluated with AUROC/FPR95/AUPR and not as pass@k reranking or candidate selection.

**C. Hidden-state vs KV embeddings; sanity checks; random-chance baselines (tEgG, rW12)**

* Authors provide hidden-state embedding results (meta-llama/Llama-3.1-8B-Instruct) under the same aggregation pipeline as KV embeddings on selected MTEB classification tasks, reporting similar performance.
* They add random-choice baselines for these tasks and argue the gap vs random supports that the pipeline is not degenerate.

**D. Token-wise vs layer-wise KV-CoE; why token dimension is used (tEgG, PbPH, 2n9n)**

* Authors directly address the key conceptual concern that KV cache is not a drop-in replacement for hidden states along layers.
* They provide an ablation showing that layer-wise KV-CoE performs substantially worse on MATH and TheoremQA, motivating the redesign of KV-CoE to operate along the token dimension, aligned with KV’s token-centric semantics.
* They also summarize pooling/aggregation strategies used for KV-CoE vs switching.

**E. Fast/slow thinking induction and implementation details (Zzrt)**

* Authors specify that fast/slow thinking is controlled purely by prompting using <think> blocks (fast: empty <think></think>, slow: open <think>), and that the KV-based controller selects between these prompts rather than modifying KV content.

**F. Hyperparameter selection transparency and “cherry-picking” concern (rW12)**

* Authors clarify a stable default selection rule described in Appendix B (KV-Classification: 2 layers, length 128; KV-Generative: 8 layers, length 32), with one stated exception (DeepSeek-14B on MATH500 with KV-Generative). They commit to explicitly stating this rule to avoid impressions of cherry-picking.

**G. Applicability beyond reasoning tasks (rW12)**

* Authors provide evidence for non-reasoning generative tasks via TruthfulQA: both KV-Classification and KV-Generative select fast thinking in 100% of cases, yielding token savings without loss in quality (as stated in the rebuttal).

**H. Difficulty label distribution and token truncation accounting (rW12)**

* Authors provide the difficulty label distribution used to train the switching model (counts over 7,500 examples), and clarify that token averages include sequences hitting the generation limit, but that this is rare on GSM8K/MATH500.

**I. Baselines/related work and reuse of reported numbers (tEgG, PbPH, Zzrt)**

* Authors clarify they are **not aiming for SOTA accuracy**, but for demonstrating KV cache as a low-cost embedding substrate; they will strengthen positioning in the revision.
* They acknowledge reuse of CoE baselines from prior work due to computational constraints, note model mismatch (LLaMA 3 vs 3.1) as a limitation, and commit to labeling these explicitly as reported baselines.
* They commit to adding missing cache-tuning related work: Prefix-Tuning and Cartridges.

**J. Minor presentation issues (PbPH, 2n9n)**

* Authors commit to fixing table caption placement, typos, metric labeling (e.g., “Accuracy” in Table 1), and formatting glitches.

## Still outstanding (but do not block acceptance in current framing)

* **Broader reasoning benchmarks and larger models (rW12, PbPH):** reviewers requested more challenging tasks (e.g., AIME/BRUMO) and larger models (e.g., Qwen3-32B). Authors acknowledge GSM8K/MATH500 may not fully stress modern reasoning models and state that additional experiments are in progress and will be included in future versions.
* **Hidden-state-based switching baseline (tEgG, Zzrt, rW12):** authors explicitly state this is conceptually possible but not the focus; the contribution is demonstrating KV-only sufficiency for switching under minimal overhead. This remains an unaddressed baseline request but is consistent with the paper’s stated scope.
* **Full re-evaluation of CoE baselines under identical model versions (PbPH, tEgG):** authors note computational constraints and model mismatch, and plan partial re-evaluations where feasible000.

**Reviewer Scores:**

* **Reviewer PbPH (original: 6)**: Likely **unchanged at 6**. The reviewer was already inclined toward acceptance; rebuttal clarifies the “two methods” concern, scope limitation around “certain classification tasks,” baseline mismatch explanation (LLaMA 3 vs 3.1), and answers token-vs-layer rationale with explicit ablations.
* **Reviewer 2n9n (original: 6)**: Likely **unchanged at 6**. The rebuttal directly answers metric clarification (MaxProb/PPL/Entropy), explains why KV cannot track layer-wise hidden state behavior and motivates token-wise redesign, provides pooling details, and states the efficiency–accuracy trade-off for switching.
* **Reviewer Zzrt (original: 4)**: Likely **remain unchanged or increase to 6**. The rebuttal addresses the key questions: CoE usage as confidence estimation (not reranking), explicit fast/slow prompting details, agreement on hidden-state optimization but clarifying scope, and commits to adding Prefix-Tuning/Cartridges. The remaining concern (missing hidden-state switching baseline) is acknowledged but scoped out.
* **Reviewer rW12 (original: 4)**: Likely **remain unchanged or increase to 6**. The rebuttal provides random chance accuracy baselines, difficulty-label distribution, explicit hyperparameter selection rule (reducing cherry-picking concern), applicability to a non-reasoning task (TruthfulQA), and clarifies token truncation inclusion. Remaining request for harder benchmarks is acknowledged as future work.
* **Reviewer tEgG (original: 4)**: Likely **remain unchanged or increase to 6**. The rebuttal addresses core technical questions: hidden-state embedding comparison with sanity checks, CoE usage clarification, a concrete token-vs-layer ablation that motivates token-wise KV-CoE, and clarifies scope vs SOTA. The primary remaining item is major writing polish (which authors explicitly commit to).

---

### Decision · Program_Chairs · 2026-01-26

Accept (Poster)